# Room Temperature In-Situ Synthesis of Inorganic Lead Halide Perovskite Nanocrystals Sol Using Ultraviolet Polymerized Acrylic Monomers as Solvent and Their Composites with High Stability

**Ludan Zhu, Shuanglong Yuan \***[ID]**, Jun Cheng, Long Chen, Chuanqi Liu, Hua Tong, Huidan Zeng and Qiling Cheng**[ID]

Institute of Inorganic Materials, East China University of Science and Technology, Shanghai 200237, China; 15261805723@163.com (L.Z.); Y30170392@mail.ecust.edu.cn (J.C.); Y30180381@mail.ecust.edu.cn (L.C.); Y30180431@mail.ecust.edu.cn (C.L.); tonghua@ecust.edu.cn (H.T.); hdzeng@ecust.edu.cn (H.Z.); chengql@ecust.edu.cn (Q.C.)

**\*** Correspondence: Shuanglong@ecust.edu.cn; Tel.: +86-21-6425-3395; Fax: +86-21-6425-3395

**Abstract:** As a kind of promising optoelectrical material, all-inorganic perovskite nanocrystals $CsPbX_3$ (X = Cl, Br, I) have attracted much attention, due to their excellent optoelectrical characteristics, in recent years. However, their synthesis approaches require rigorous conditions, including high temperature, eco-unfriendly solvent or complex post-synthesis process. Herein, to overcome these issues, we reported a novel facile room temperature in-situ strategy using ultraviolet polymerizable acrylic monomer as solvent to synthesis $CsPbX_3$ nanocrystals without a complex post-synthesis process. In this strategy, adequate soluble precursors of Cs, Pb and Br and reaction terminating agent 3-aminopropyltriethoxysilane (APTES) were used. The obtained $CsPbBr_3$ nanocrystals showed a high photoluminescence quantum yields (PLQY) of 87.5%. The corresponding polymer composites, by adding light initiator and oligomer under ultraviolet light radiation, performed excellent stability in light, air, moisture and high temperature. The reaction process and the effect of the reaction terminating agent have been investigated in detail. This strategy is a universal one for synthesizing $CsPbX_3$ nanocrystals covering visible light range by introducing HCl and $ZnI_2$.

**Keywords:** perovskite nanocrystals; acrylic monomers; in-situ synthesis

## 1. Introduction

In recent years, inorganic lead halide perovskite $CsPbX_3$ (X = Cl, Br, I) nanocrystals have drawn attention because of their potential applications in light emitting diodes (LEDs)**,** solar cells, lasers, fluorescent probes, display backlights et al. [1–13], owing to their excellent optoelectrical properties [1–3], such as high photoluminescence quantum yield (PLQY), tunable emission spectra over the whole visible spectrum (400–700 nm), relatively narrow full width at half maximum (FWHM) (12–42 nm), facile solution synthetic process and low cost [4,14,15].

At present, many synthetic approaches have emerged to prepare the high performance of perovskite nanocrystals (PNCs) and their composites, including hot injection [4], supersaturated recrystallization [15], top-down [16], mechanochemistry [17], in-situ synthesis [18], microwave assistant [19], volatilization of solvent [20], etc. However, high temperature, gas protection, eco-unfriendly solvent and other conditions required by these methods lead to difficulty for mass production [4,12]. For instance, the employed solvents include N, N-dimethylformamide (DMF) [2], toluene [15], 1-Octadecene (ODE) [21] and dimethyl sulfoxide (DMSO) [22], because of the need to

dissolve Cs, Pb and halogen precursors. To solve this problem, many researchers have devoted time to exploring an eco-friendly and cost-effective strategy, such as aqueous or alcohol synthesis [23–27]. Recently, Li et al. firstly achieved direct synthesis of $CsPbBr_3$ nanorods via tip ultrasonication by using a series of short chain alcohols, except for ethanol [28], offering a novel synthetic strategy in polar solvent. Nevertheless, this strategy still needs post-synthesis treatment and redispersion in toluene.

Many efforts have been devoted into the color-conversion perovskite nanocrystals-polymer composite film applied in liquid crystal display backlight in recent years. One typical strategy is to blend perovskite nanocrystals sols synthesized via different approaches with polymer, then the composite film can be obtained by solvent volatilization. The mixing of perovskite nanocrystals with polymers often causes a predictable agglomeration of nanocrystals, due to the large difference in polarity between perovskite nanocrystals and polymers [29]. To avoid the aggregation of perovskite nanocrystals in polymers, an in-situ synthetic strategy, which firstly adopts an appropriate solvent to dissolve Cs, Pb, Br sources and polymer, and then follows a solvent volatilization process, has been reported [14,30]. For instance, Zhong et al. reported an in-situ synthesis of perovskite nanocrystals-polyvinylidene fluoride (PVDF) composite film by employing DMF as solvent of CsBr, $PbBr_2$ and PVDF [14]. Chen et al. realized an in-situ large-scale continuous production of PNCs embedded in polymethyl methacrylate (PMMA) via microfluidic spinning microreactors, by using chloroform as solvent and Cs-oleate, Pb-oleate and tetraoctylammonium bromide as precursors [30]. Dong et al. developed an in situ polymer swelling/deswelling strategy to fabricate $MAPbBr_3$/polystyrene and $MAPbBr_3$/polycarbonate composite films [31]. Moreover, these hydrophobic polymers tend to form a coherent barrier layer around perovskite nanocrystals, remarkably enhancing the stability of perovskite nanocrystals against moisture and oxygen. However, these approaches still inevitably employ eco-unfriendly solvent. Recently, Tong et al. developed a modified hot-injection synthesis of $CsPbBr_3$ nanocrystals by using ultraviolet (UV)-polymerizable lauryl methacrylate (LMA) as a solvent, which effectively diminished the use of consumptive solvent, opening a new window to synthesize $CsPbX_3$ polymer composite films [29]. Regretfully, this strategy still needs high reaction temperature (180 °C) and an ice bath to terminate the reaction.

Herein, we developed a facile in-situ one-pot synthesis of $CsPbX_3$ nanocrystals using acrylic monomer as solvent at room temperature. Unlike the high temperature hot-injection method in a kind of solvent [4] and the supersaturated recrystallization method using two kinds of solvent [15], two key points have to be considered before the synthesis of $CsPbX_3$ nanocrystals in a kind of solvent at room temperature. On one hand, adequate soluble precursors of Cs, Pb and Br should be included in the solvent. On the other hand, the growth of perovskite crystals owing to the desorption of conventional ligands, such as oleylammonium and oleic acid, should be restricted [32]. Therefore, in this work, we employed Cs oleate, Pb oleate and tetraoctylammonium bromide as precursors, because of their high enough solubility in acrylic monomer. In addition, we introduced APTES as ligands to restrict the crystal growth of $CsPbX_3$ nanocrystals, due to the strong chemical interaction between the $-NH_2$ group and $CsPbX_3$ [33–36]. APTES, which usually acts as the precursors of $SiO_2$ coating [37,38], and also avoids the desorption of ligands, owing to the cross-linking Si-O-Si network around $CsPbX_3$ by hydrolysis condensation. The synthesis of $CsPbBr_3$ nanocrystals showed a high PLQY of 87.5%, as well as excellent stability for corresponding film in light, air, moisture and high temperature. The present strategy has the following advantages: (1) direct one-pot synthesis of $CsPbX_3$ sol at room temperature without complex post-synthesis process, such as purification, centrifuge and redispersion; (2) easy for mass production even fabrication online for film by an ultraviolet (UV) curing technology; (3) uniformly distributed $CsPbBr_3$ nanocrystals within the polymer matrix, compared to the inevitable aggregation in blending method. Moreover, this strategy can be expanded to the synthesis of red ($CsPbI_3$) and blue ($CsPbCl_3$) nanocrystals by introducing an appropriate amount of $ZnI_2$ and HCl, respectively.

## 2. Experiment

### 2.1. Materials

All raw materials, including Cesium carbonate ($Cs_2CO_3$, 99.9%,), lead oxide (PbO, 99.0%), oleic acid (OA, 90%), ethanol (EtOH,), tetraoctylammonium bromide (TOAB, 98%), 2-hydroxy-2-methylpropiophenone ($C_{10}H_{12}O_2$, 97%), isobornyl methacrylate (IBOMA, AR), isobornyl acrylate (IBOA, AR), lauryl methacrylate (LMA, AR), methyl mtethacrylate (MMA, AR), 3-aminopropyltriethoxysilane (APTES, 99.0%), hydrogen chloride (HCl, 99.0%) and zinc iodide ($ZnI_2$, 99.0%), were purchased from Sigma-Aldrich (Sigma-Aldrich Company, Shanghai, China) and used directly without further purification.

### 2.2. Synthesis of CsPbBr3 Sol/Composites

$Cs_2CO_3$ (0.5 mmol, Aldrich, 99.9%), PbO (1 mmol, Aldrich, 99.0%), OA (2.5 mL, Sigma-Aldrich, 90%) and acrylic monomers (IBOMA/IBOA/LMA/MMA, 45 mL) were added into a 100 mL 3-neck flask and then heated to 75 °C. After fully reacting, the solution was cooled to room-temperature for preparing $CsPbBr_3$ nanocrystals.

Acrylic monomers (IBOMA/IBOA/LMA/MMA, 20 mL), OA (1 mL) and TOAB (0.3 mmol) were added into a 100 mL beaker to obtain a TOAB solution. The TOAB solution was swiftly injected into above prepared Cs-oleate and Pb-oleate solution (5 mL) and 60 s later APTES (0.3 mmol) was injected. It was stirred for 4 h in air and then preserved in the reagent bottle for further use.

For the preparation of block, blended 25 mL $CsPbBr_3$ solution (prepared in IBOMA) with light initiator 2-hydroxy-2-methyl-1-acetone (0.58 g) and then added them into a 50 mL beaker. After stirred for 5 min, the solution was poured into a template and then cured by a 395 nm UV lamp. The thickness of the block can be changed according to depth of the template. For the preparation of film, 25 mL $CsPbBr_3$ solution (prepared in IBOMA), 0.58 g light initiator 2-hydroxy-2-methyl-1-acetone and 75 g oligomer have been used.

### 2.3. Anion-Exchange Process

Certain amounts of HCl or $ZnI_2$ (Table 1) as the anion source were mixed with IBOMA (10 mL) in a beaker, and stirred until they were dissolved completely. When TOAB-IBOMA solution was injected into the Cs-oleate and Pb-oleate IBOMA solution, the $CsPbBr_3$ nanocrystals were obtained, and the sol showed bright green under the ultraviolet light (395 nm). The HCl/$ZnI_2$-IBOMA solution was added into $CsPbBr_3$ sol, and 60 s later the APTES (0.3 mmol) was injected. After stirring for 4 h in air, the anion-exchanged samples were obtained.

**Table 1.** Preparation data of $CsPbX_3$ (X = Cl, Br, I) nanocrystals samples.

| Cl:Br | | | | | | |
|---|---|---|---|---|---|---|
| Ratio | 1.2 | 1.0 | 0.8 | 0.65 | 0.5 | 0.35 | 0 |
| HCL (µL) | 31.5 | 27 | 22.5 | 18 | 13.5 | 9 | 0 |

| I:Br | | | | | | |
|---|---|---|---|---|---|---|
| Ratio | 0 | 0.35 | 0.5 | 0.65 | 0.8 | 1.0 | 1.2 |
| $ZnI_2$ (g) | 0 | 0.016 | 0.024 | 0.032 | 0.04 | 0.048 | 0.056 |

### 2.4. Characterization Methods

UV-VIS absorption spectra were collected using a 950 UV-Visible Spectrophotometer (Shimadzu Corporation, Kyoto, Japan) with absorption mode. Photoluminescence (PL) spectra were measured by a Fluorolog-3-P UV-VIS-NIR fluorescence spectrophotometer (Jobin Yvon, Longjumeau, France) with a 450 W Xe lamp as the excitation source. PLQY was estimated according to standard procedure, using Rhodamine B (95% of the yield in ethanol) as reference. For samples and rhodamine ethanol

solution, an excitation wavelength of 450 nm was used. Powder X-ray diffraction (XRD) patterns were collected using a Bruker AXS D8 Advance diffractometer (Bruker AXS Ltd., Karlsruhe, Germany) with a step width of 0.02° (Voltage 50 kV, current 40 mA, Cu-Ka). The $CsPbBr_3$ acrylic sols (after adding APTES) were precipitated by adding 1-butyllactone, and then centrifuged in a centrifuge for 10 min (10,000 r/min). The precipitate was freeze-dried and ground to produce the yellow powder samples for XRD test. The morphology and particles size of the samples were examined by a JEM-2100 (JEOL Ltd.,Tokyo, Japan) transmission electron microscopy (TEM). High-resolution TEM (HRTEM) images were recorded using a JEOL JEM-2100 microscope (JEOL Ltd., Tokyo, Japan) operated at 200 kV. After adding 2-hydroxy-2-methylpropiophenone (0.58 g) as a photoinitiator, $CsPbBr_3$ blocks were prepared via an ultraviaolet polymerization method. The $CsPbBr_3$ composite sample for TEM measurement was obtained by means of ultrathin frozen section. Fourier transform infrared (FTIR) spectra in the region of 400~4000 cm$^{-1}$ were recorded on a Thermo Fisher Scientific Nicolet 6700 Spectrometer (Thermo Nicolet Corporation, Basingstoke, United Kindom) by pressing the mixture of the powder sample (obtained after centrifugation and freeze-drying) and KBr to a sheet.

## 3. Results and Discussion

As schematically shown in Figure 1a, the synthetic process was performed in two steps: the formation of $CsPbBr_3$ nanocrystals and size controlling by adding APTES. Details of the process have been described in the Experimental section. The sol sample showed bright green under the ultraviolet light (395 nm) (Figure 1c), and $CsPbBr_3$ nanocrystals/polyarcylic composites can be prepared via an ultraviaolet polymerization method (Figure 1b,c).

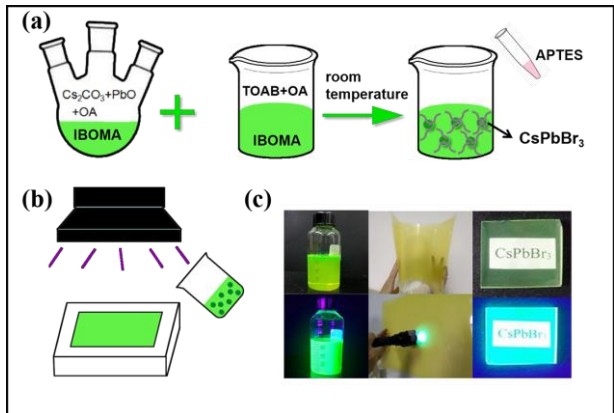

**Figure 1.** (**a**) In-situ one-step synthesis process of $CsPbBr_3$ nanocrystals sol; (**b**) preparation of solid $CsPbBr_3$ nanocrystals composites; (**c**) nanocrystals sol, blocks and films.

Figure 2a shows the absorption and fluorescence spectra of $CsPbBr_3$ nanocrystals prepared in IBOMA. An emission peak at 510 nm was observed, and the corresponding absorption peak located at 485 nm. According to the formula proposed by Tauc and Mott, the calculated bandgap of as-prepared $CsPbBr_3$ nanocrystals is 2.25 eV [39,40]. However, red shift can be observed in the emission spectra of $CsPbBr_3$ prepared in other three acrylic monomers, indicating the particle size increasing [41] (Figure S1 in Supplementary Materials). It was noted that the PLQY of $CsPbBr_3$ nanocrystals prepared in IBOMA reached 87.5%, much higher than the samples prepared in other acrylic monomers (Table S1 in Supplementary Materials). In order to study the crystal structure of the samples, the XRD patterns of $CsPbBr_3$ powder samples are shown in Figure 2b and Figure S1, and $CsPbBr_3$ nanocrystals possessing a cubic structure (PDF # 54-0752) were synthesized in IBOMA. However, a small impurity peak (Figure S1) identified as $PbBr_2$ existing in the pattern of $CsPbBr_3$ nanocrystals synthesized in IBOA. The XRD results demonstrated that cubic $CsPbBr_3$ nanocrystals can be synthesized at room temperature by this method.

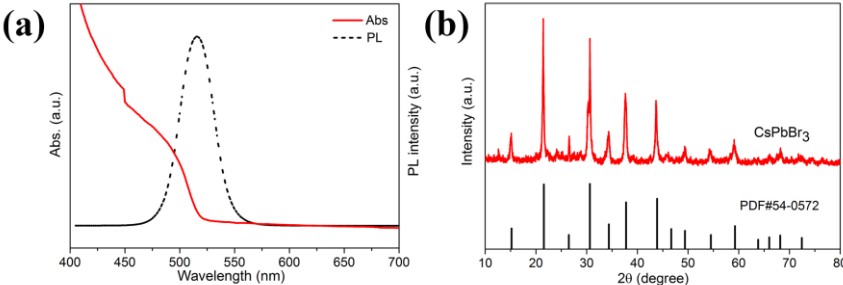

**Figure 2.** (**a**) Photoluminescence (PL) (dark dash) and absorption (red line) spectra, and; (**b**) X-ray diffraction (XRD) pattern of CsPbBr$_3$ nanocrystals synthesized in isobornyl methacrylate (IBOMA).

TEM images of the as-synthesized CsPbBr$_3$ nanocrystals are shown in Figure 3. Compared to other three samples (Figure S2 in Supplementary Materials), CsPbBr$_3$ synthesized in IBOMA (Figure 3a) has a smaller particle size about 8.4 ± 1.3 nm (Figure 3b). The HRTEM image (Figure 3c) reveals high crystallinity with a lattice spacing of 0.295 nm, which fits well with the (200) plane of cubic CsPbBr$_3$. According to the TEM results and the PLQY performance, CsPbBr$_3$-IBOMA sol has been used for the subsequent studies and tests.

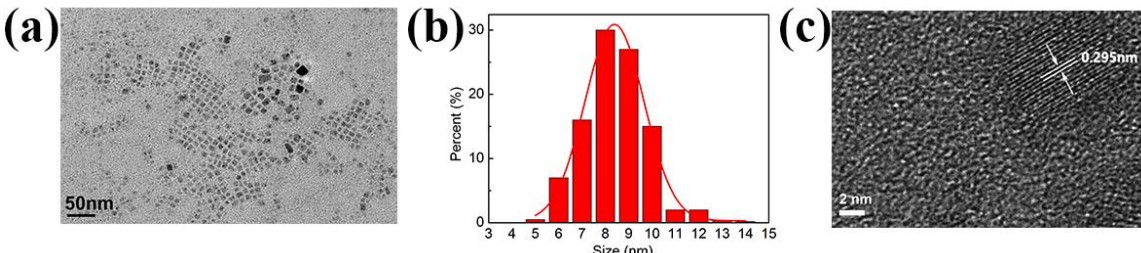

**Figure 3.** (**a**) Transmission electron microscopy (TEM) image; (**b**) size distribution, and; (**c**) high-resolution TEM (HRTEM) with crystalline interplanar spacing of CsPbBr$_3$ nanocrystals in IBOMA.

The CsPbBr$_3$-IBOMA nanocrystals sol exhibited excellent stability. After being placed in air without any protection for 2 weeks, the particle size remained nearly unchanged (Figure 4a), which was attributed to the capping of hydrolyzed APTES ligand [32]. APTES hydrolyzes on the surface of nanocrystals, forming a network structure to prevent the desorption of ligands and to subsequently prohibit size increases, which enhanced the stability of nanocrystals [37,42]. To further prove the effect of APTES, CsPbBr$_3$-IBOMA nanocrystals without APTES were synthesized, and their morphology and size were determined by TEM (Figure 4b). The size (45.7 ± 7.7 nm) was obviously larger than the sample using APTES (8.4 ± 1.3 nm). The reaction with and without APTES revealed a large difference (Video S1). Because of unrestricted increasing of particle size, the APTES-free sample rapidly turned to yellow after mixing two precursors, and finally the sample subsided [43]. Meanwhile, the sample with APTES remained unchanged and no aggregation was observed, suggesting that the addition of APTES effectively prevents the size increasing of CsPbBr$_3$ nanocrystals. FTIR spectrum was used to demonstrate the capping of APTES, as shown in Figure 4c, in which it exists a N-H vibration absorption peak at 3448 cm$^{-1}$, originating from the -NH$_2$ group of APTES. The peak at 948 cm$^{-1}$ was ascribed to the vibration absorption of the -Si-OH group of hydrolyzed APTES. The peak at 1125 cm$^{-1}$ originated from the vibration absorption of the -Si-O-Si- group, owing to the hydrolysis condensation between the -Si-OH groups or the -Si-OH and -SiOC$_2$H$_5$ groups [37]. Therefore, the mechanism of CsPbBr$_3$ nanocrystals with APTES can be summarized as follows: the polar -NH$_2$ group of APTES closely connects to CsPbBr$_3$ nanocrystals and three ethoxyl siloxane groups hydrolyze to form -Si-OH, following a dehydration process between the -SiOH groups or -SiOH and -SiOC$_2$H$_5$, resulting in the formation of an Si-O-Si cross-linking network structure (Figure 4d). In the whole process, a small amount of water in air and solvent advanced the hydrolysis reaction slowly [44].

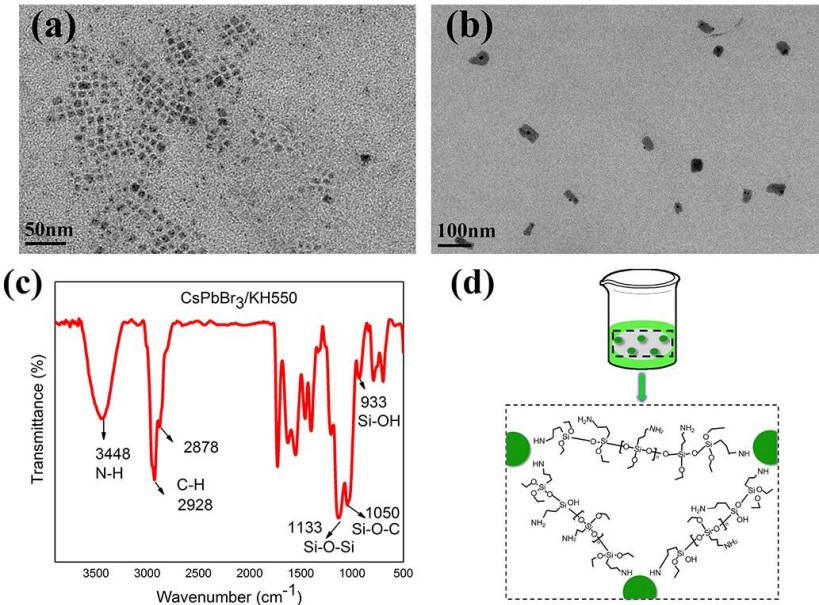

**Figure 4.** TEM images of CsPbBr$_3$ nanocrystals in IBOMA (**a**) with 3-aminopropyltriethoxysilane (APTES) and (**b**) without APTES after 2 weeks; (**c**) Fourier transform infrared (FTIR) spectrum of CsPbBr$_3$ nanocrystals in IBOMA; (**d**) illustration of formation of Si-O-Si cross-linking network.

The present strategy can be expanded to synthesize blue and red emission CsPbX$_3$ by adding HCl or ZnI$_2$ to adjust the ratio of halide ions. Details of the process have been described in the Experimental section. The XRD patterns of the CsPbBr$_3$ nanocrystals and anion-exchanged samples showed the retention of cubic perovskite structure, and the shift of the XRD reflections was observed due to the difference of anion radius (Figure S3 in Supplementary Materials). Their PL spectra were adjustable from 430 to 620 nm (Figure S4 in Supplementary Materials) with narrow FWHM (10−40 nm), indicating this is a universal strategy.

It is well known that the high sensitivity to the oxygen, moisture, light and temperature etc. leads to the poor stability of perovskites nanocrystals [13,45,46]. In order to enhance the stability for further application, in previous work, CsPbBr$_3$ nanocrystals were often combined with polymer by complicated processing [12]. Herein, by a facile UV curing process, CsPbBr$_3$-IBOMA sol, can be cured directly into various shapes, because of the polymerization of the IBOMA monomer. The morphology of the CsPbBr$_3$ nanocrystals block sample was studied by TEM. As shown in Figure 5a, the CsPbBr$_3$ nanocrystals were uniformly distributed in the polymer matrix without phase separation or aggregation as expected, owing to the in-situ method. The particle size (11.6 ± 2.2 nm) showed a slight increase compared with the sample before curing, because of the UV irradiation [47]. Furthermore, the emission spectrum was also unchanged, and only the emission intensity slightly reduced (Figure 5b), suggesting that the polymerization process of IBOMA does not affect the performance of nanocrystals.

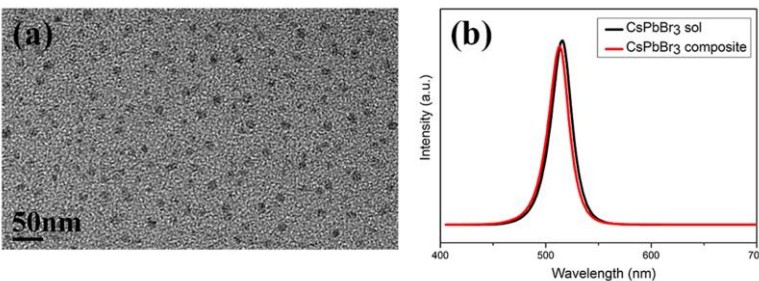

**Figure 5.** (**a**) TEM image of the CsPbBr$_3$ nanocrystals block sample; (**b**) fluorescence spectra of samples before and after be cured.

High stability is very crucial for the practical application of CsPbBr$_3$ nanocrystals. Therefore, the light, air, moisture and thermal stability of as-synthesized nanocrystals were investigated. The PL spectra of the CsPbBr$_3$-IBOMA block sample was measured at 5 h intervals under continuous irradiation with 395 nm UV LEDs (3W). As shown in Figure 6a, the shape and peak position of the emission spectra did not change significantly with the increase of irradiation time, indicating that the particle size of CsPbBr$_3$ nanocrystals was not affected by UV irradiation. However, the luminous intensity of the sample showed a decreasing trend (Figure 6b). It increased in the initial 10 h, and then went through slight decreasing. Finally, it decreased only by 9.7%, to 0.903 after 70 h. The results exhibited the excellent light stability of CsPbBr$_3$ nanocrystals, due to the encapsulation of IBOMA and the coating of APTES.

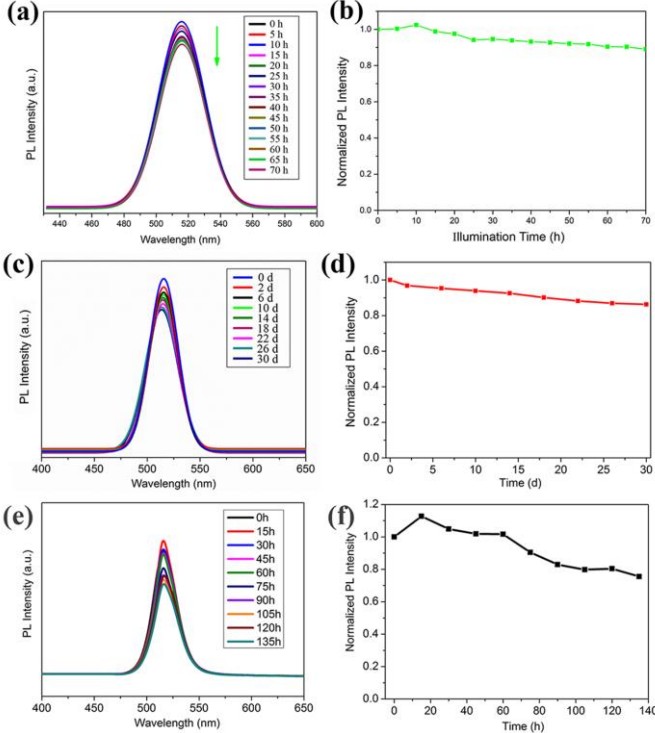

**Figure 6.** PL spectra and intensity attenuation of CsPbBr$_3$-IBOMA block sample (**a**,**b**) at 5 h intervals under 395 nm UV LEDs; (**c**,**d**) in air at 4-day intervals; (**e**,**f**) in water at 15 h intervals. (All samples were excitated under 395 nm light).

To explore the stability in air, the CsPbBr$_3$-IBOMA nanocrystals block sample was placed in the air without any protection and the emission spectrum was measured every 4 days under laboratory conditions (K = 298 K, 57% RH). The emission peak of the CsPbBr$_3$ sample kept at about 510 nm steadily without any shift (Figure 6c), indicating unchanged particle size. In addition, the fluorescence intensity maintained a high level of 89.5% at 30 days (Figure 6d), compared with the previous work [48].

To evaluate the water and thermal stability of CsPbBr$_3$ nanocrystals, we kept the CsPbBr$_3$-IBOMA nanocrystals block sample immersed in water and heated to 348.15 K. The fluorescence intensity of the emission spectrum was tested at 15 h intervals. As shown in Figure 6e, the emission spectra shape and peak position of the sample nearly did not change, while the luminous intensity decreased gradually (Figure 6f). Within the initial 15 h, it went through a slight rise and then decreased slowly. Finally, after 135 h, the luminous intensity reached 75.6%. The result is comparable with previous reported CsPbBr$_3$ nanocrystals [29].

## 4. Conclusions

In summary, we have introduced a controllable one step in-situ preparation method at room temperature in acrylic monomer. The cubic phase $CsPbBr_3$ nanocrystals with outstanding luminous performance have been successfully synthesized. The photoluminescence quantum yield is up to 87.5%. Isobornyl methacrylate as solvent can be cured directly by ultraviolet light to form transparent solid materials, such as block and film. In this way, not only can a large amount of organic waste liquid be avoided, but also the acrylic polymer encapsulates the nanocrystals and limits further growth. The $CsPbBr_3$-IBOMA nanocrystals block showed great stability under the conditions of air, continuous irradiation and water. In addition, the ligands APTES adsorbing on the surface of $CsPbBr_3$ nanocrystals effectively controlled its particle size at $8.4 \pm 1.3$ nm, which ensured its excellent luminous performance.

**Supplementary Materials:** The following are available online at http://www.mdpi.com/2076-3417/10/9/3325/s1, Figure S1. (**a**) PL and Abs spectra and (**b**) XRD patterns of $CsPbBr_3$ nanocrystals synthesized in different acrylic monomers, Table S1. Photoluminescence quantum yield of $CsPbBr_3$ in different acrylic monomers, Figure S2. TEM images of nanocrystals synthesized (**a**) in IBOA ($61.5 \pm 5.8$ nm), (**b**) in MMA ($40.5 \pm 8.6$ nm), (**c**) in LMA ($15.2 \pm 1.8$ nm), Figure S3. XRD patterns of the $CsPbBr_3$ nanocrystals and anion-exchanged samples, Figure S4. Emission spectra of $CsPbX_3$ (X = Cl, Br, I) nanocrystals, Figure S5. Photograph of perovskite $CsPbX_3$ (X = Cl, Br, I) nanocrystals. Video S1: comparison between samples with and without APTES.

**Author Contributions:** L.Z. prepared the samples and wrote this manuscript. J.C., C.L. and L.C. processed and analyzed the signal data. S.Y., H.T., H.Z., and Q.C. conceived the research and provided many suggestions for the research method and revised the article completely. All authors have read and agreed to the published version of the manuscript.

**Funding:** This research received no external funding.

**Conflicts of Interest:** There is no conflict to declare.

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
