# Peer review of "Room Temperature In-Situ Synthesis of Inorganic Lead Halide Perovskite Nanocrystals Sol Using Ultraviolet Polymerized Acrylic Monomers as Solvent and Their Composites with High Stability"

_applsci, doi:10.3390/app10093325_

Round 1

Reviewer 1 Report

The manuscript by Zhu et al. is devoted to the synthesis of perovskite nanocrystals with improved stability. The authors provided a detailed experimental analysis of the synthesized nanocrystals. The most exciting result is the fabrication of composite based on synthesized nanocrystals which showed high stability in ambient conditions and even when dipped in water. Although the results are novel and in demand for the development of the photonic applications, the manuscript needs major revision according to the comments provided below. After addressing those issues, the manuscript may be published in Applied Sciences.

Major issues

  1. The major drawback of this manuscript is its poor English and lack of logic in the story.
  2. Authors claimed that the synthesis of PNC is room-temperature; however, they heated the precursors up to 75C before crystallization. In addition, I suggest merging the 2.2-2.4 paragraphs. Please specify: what acrylic monomers are used in the preparation of composite. Also, be consistent in the use of abbreviations.
  3. I disagree that in the 2.5 section an anion-exchange process is described. This is just a synthetic procedure to form perovskite NC with different Cl:Br and I:Br ratio. So, the discussion provided further (in “ Results and discussion” section, Page 6) is quite confusing.
  4. Another issue in the Experiment section is that there is no correct designation of samples, which results in confusion throughout the text of the manuscript. I suggest the authors think about the sample names.

Minor issues

  1. Please specify what reaction terminating agent did you use in the abstract
  2. Please specify each reference work to each property in “At present, many synthetic approaches have emerged to prepare a high performance of 36 perovskite nanocrystals (PNCs) and their composites, including hot injection, supersaturated recrystallization, top-down, mechanochemistry, in-situ synthesis, microwave assistant, volatilization of solvent[4,14-18], etc.”
  3. Lots of the text should be rephrased. For example, this sentence is quite hard to understand “These approaches need either high temperature, inert gas 39 protection, high boiling point, and eco-unfriendly solvent, or low PLQY, or complex post-synthesis 40 treatments, leading to difficulty for mass production”.
  4. In the introduction section, where authors should introduce their findings, the text is mixed with literature review. I suggest relocating this part (Page 2, lines 73-87) to the discussion of recent works.
  5. In the Materials section, some of the chemicals are listed starting from capital letters, and others are not. Be consistent.
  6. In “Characterization methods”: Provide the name of the manufacturer of 950 UV-Visible Spectrophotometer. What type of Rhodamine did you use? If it was 6G, it has very low absorption at 450 nm, and data may be incorrect. What CsPbBr3 solvents did you use for XRD measurements (Mixed with acrylic monomers or with APTES or final solution with C10H12O2)? What is the CsPbBr3 block sample used for TEM measurement?
  7. In “Results and discussion” section authors started with the description of the synthesis and gave more details in this section compared to that given in “Experimental”.
  8. Page 4, lines 161-163: “However, red shift can be observed in the emission spectra of CsPbBr3 prepared in other three solvents, indicating the particle size increasing[41] (Figure S1)”. As far as I understood the type of acrylic monomers was changed – not the solvent.
  9. Page 4, lines 165-168: the sample preparation procedure for XRD was already stated in the “Experimental” section.
  10. Particle size deviation is wrong; please fit the data with a normal distribution. It should be twice less.
  11. Page 5, line 184: “After being placed for 2 weeks <…>” Specify the conditions of storage.
  12. Page 5, lines 195-196: as abovementioned, there is no need to repeat the formation procedure of the sample for other measurements.

Author Response

Response to Reviewer 1 Comments

Point 1: The major drawback of this manuscript is its poor English and lack of logic in the story.

Response 1: We are sorry for language and logic problems. The whole manuscript has been examined carefully and revised some grammar problems .

Point 2: Authors claimed that the synthesis of PNC is room-temperature; however, they heated the precursors up to 75℃ before crystallization.

Response 2: Thanks to the reviewer’s comment. The precursors were heated to accelerate dissolution of Cs2CO3 and PbO. Appropriate heating temperature is only for shortenning the time of complete dissolution.

Point 3: In addition, I suggest merging the 2.2-2.4 paragraphs.

Response 3: Thanks to the reviewer’s advice. 2.2-2.4 paragraphs have been merged in 2.2.

Point 4: Please specify: what acrylic monomers are used in the preparation of composite. Also, be consistent in the use of abbreviations.

Response 4: Thanks to the reviewer’s comment. Acrylic monomer used in the preparation of composite has been added in the manuscript.

Point 5: I disagree that in the 2.5 section an anion-exchange process is described. This is just a synthetic procedure to form perovskite NC with different Cl:Br and I:Br ratio. So, the discussion provided further (in “ Results and discussion” section, Page 6) is quite confusing.

Response 5: Thanks to the reviewer’s comment. The anion exchange often refers to the exchange of halide ions in the as-prepared CsPbX3 NCs ( ACS Appl. Mater. Interfaces 2019, 11, 14256−14265,line 15-16).In this manuscript, HCl or ZnI2 as the anion source were added after the TOAB solution was injected into Cs-oleate and Pb-oleate solution(At that time, the CsPbBr3 nanocrystals was obtained) .So the process of ion exchange has happened. We are sorry for unclear expression, and the description has been revised in detail.

Point 6: Another issue in the Experiment section is that there is no correct designation of samples, which results in confusion throughout the text of the manuscript. I suggest the authors think about the sample names.

Response 6: Thanks to the reviewer’s advice. The samples have been distinguished in naming in order to reduce confusion.

Point 7: Please specify what reaction terminating agent did you use in the abstract.

Response 7: Thanks to the reviewer’s advice. Terminating agent has been added in the abstract.

Point 8: Please specify each reference work to each property in “At present, many synthetic approaches have emerged to prepare a high performance of 36 perovskite nanocrystals (PNCs) and their composites, including hot injection, supersaturated recrystallization, top-down, mechanochemistry, in-situ synthesis, microwave assistant, volatilization of solvent[4,14-18], etc.”

Response 8: Thanks to the reviewer’s advice. References have been detailed in this sentence.

Point 9: Lots of the text should be rephrased. For example, this sentence is quite hard to understand “These approaches need either high temperature, inert gas 39 protection, high boiling point, and eco-unfriendly solvent, or low PLQY, or complex post-synthesis 40 treatments, leading to difficulty for mass production”.

Response 9: Thanks to the reviewer’s advice. We checked the manuscript carfully and some sentences have been revised, as shown in the manuscript with annotation.

Point 10: In the introduction section, where authors should introduce their findings, the text is mixed with literature review. I suggest relocating this part (Page 2, lines 73-87) to the discussion of recent works.

Response 10: Thanks to the reviewer’s advice. This literature review has been removed from the original part.

Point 11: In the Materials section, some of the chemicals are listed starting from capital letters, and others are not. Be consistent.

Response 11: Thanks to the reviewer’s advice. All the chemicals are listed starting from lower-case letters.

Point 12: In “Characterization methods”: Provide the name of the manufacturer of 950 UV-Visible Spectrophotometer. What type of Rhodamine did you use? If it was 6G, it has very low absorption at 450 nm, and data may be incorrect. What CsPbBr3 solvents did you use for XRD measurements (Mixed with acrylic monomers or with APTES or final solution with C10H12O2)? What is the CsPbBr3 block sample used for TEM measurement?.

Response 12: Thanks to the reviewer’s advice. The reference material for PLQY testing is Rhodamine B. The CsPbBr3 solvents we used for XRD measurements and the CsPbBr3 block sample used for TEM measurement have been detailed in the manuscript.

Point 13: In “Results and discussion” section authors started with the description of the synthesis and gave more details in this section compared to that given in “Experimental”.

Response 13: Thanks to the reviewer’s advice. Synthetic process description has been removed from the section 3 and we have supplemented the experimental part.

Point 14: Page 4, lines 161-163: “However, red shift can be observed in the emission spectra of CsPbBr3 prepared in other three solvents, indicating the particle size increasing[41] (Figure S1)”. As far as I understood the type of acrylic monomers was changed – not the solvent.

Response 14: Thanks to the reviewer’s advice. This sentence has been modified in the manuscript.

Point 15: Page 4, lines 165-168: the sample preparation procedure for XRD was already stated in the “Experimental” section.

Response 15: Thanks to the reviewer’s advice. The unnecessary description has been removed from the text.

Point 16: Particle size deviation is wrong; please fit the data with a normal distribution. It should be twice less.

Response 16: Thanks to the reviewer’s advice. All the size data have been fitted with a normal distribution, and the size is expressed as μ±σ  in the manuscript.

Point 17: Page 5, line 184: “After being placed for 2 weeks <…>” Specify the conditions of storage.

Response 17: Thanks to the reviewer’s advice. The conditions of storage(in air without any protection) has been specified in the text.

Point 18: Page 5, lines 195-196: as abovementioned, there is no need to repeat the formation procedure of the sample for other measurements.

Response 18: Thanks to the reviewer’s advice. The unnecessary description has been removed from the text.

Reviewer 2 Report

The paper presents a method of synthesis for inorganic lead halide perovskite CsPbBr3 at room temperature which is great.

Overall, I find the flow a little confused which makes it difficult to understand it.

Some places are lacking of citations. Such as line nr. 30-35; line nr. 36-39;

The sentence “As one of the most attractive applications …….” sounds incomplete.

Section 3. Results and discussion, started with synthetic process description which I recommend to be part of the section 2. Experiment.

Figure 3.c I find it little dark, not clear enough.

I recommend for the section for the 3. Results and discussion to include without the experimental description. The experimental description shall be put in the 2. Experiment. Example of such situations where my recommendation refers: line 219-225 belongs to experimental; line 243-245 belongs to experimental; line 249-251 belongs to experimental.

Author Response

Response to Reviewer 2 Comments

Point 1: Overall, I find the flow a little confused which makes it difficult to understand it.

Response 1: We are sorry for problems in the organization and writing of the article . The whole manuscript has been examined and revised carefully for easier understanding.

Point 2: Some places are lacking of citations. Such as line nr. 30-35; line nr. 36-39;

Response 2: Thanks to the reviewer’s comment. Citations have been added to the manuscript.

Point 3: The sentence “As one of the most attractive applications …….” sounds incomplete.

Response 3: Thanks to the reviewer’s comment. This sentence has been revised in the manuscript.

Point 4: Section 3. Results and discussion, started with synthetic process description which I recommend to be part of the section 2. Experiment.

Response 4: Thanks to the reviewer’s advice. Synthetic process description has been removed from the section 3.

Point 5: Figure 3.c I find it little dark, not clear enough.

Response 5: Thanks to the reviewer’s advice. The brightness of Figure 3c has been adjusted in the manuscript.

Point 6: I recommend for the section for the 3. Results and discussion to include without the experimental description. The experimental description shall be put in the 2. Experiment. Example of such situations where my recommendation refers: line 219-225 belongs to experimental; line 243-245 belongs to experimental; line 249-251 belongs to experimental.

Response 6: Thanks to the reviewer’s advice. These experimental descriptions have been removed from section 3 and moved to section 2.

Round 2

Reviewer 1 Report

I agree with the changes introduced to the manuscript. It can be published.

Author Response

Thanks to the reviewer’s comment.The whole manuscript has been examined and revised carefully.

Reviewer 2 Report

1.
The overall flow needs more improvement.
2.
“acrylic monomers (IBOMA/IBOA/LMA/MMA, 45 mL)” line 105 and “Acrylic monomers(IBOMA/IBOA/LMA/MMA, 20 mL)” line 108 I do not understand how the quantity, 45 ml and 20 ml, refers to each of them. I would good to specify it.
3.
My recommendation for the section 3. Results and discussion to remove the experimental description. The experimental description shall be put in the 2. Experiment. It refers to the line 146-158.

Author Response

Response to Reviewer 2 Comments

Point 1: The overall flow needs more improvement.

Response 1: We are sorry for problems in the organization and writing of the article . The whole manuscript has been examined and revised carefully.

Point 2: ”acrylic monomers (IBOMA/IBOA/LMA/MMA, 45 mL)” line 105 and “Acrylic monomers(IBOMA/IBOA/LMA/MMA, 20 mL)” line 108 I do not understand how the quantity, 45 ml and 20 ml, refers to each of them. I would good to specify it.

Response 2: Thanks to the reviewer’s comment. The obtained CsPbBr3 sol was 0.1mmol nanocrystals in 30 ml solvents. In order to avoid the errors caused by different batches of experiments, we prepared  Cs-oleate and Pb-oleate solution in one batch by expanding the amount of  Cs2CO3 and PbO by ten times and dissolving them in 50ml solvent (OA+acrylic monomers). Then 5 ml Cs-oleate and Pb-oleate solution was swiftly injected into 20 ml TOAB solution and 60 seconds later APTES (0.3 mmol) was injected. Finally, 30 ml CsPbBr3 sol was obtained.

Point 3: My recommendation for the section 3. Results and discussion to remove the experimental description. The experimental description shall be put in the 2. Experiment. It refers to the line 146-158.

Response 3: Thanks to the reviewer’s advice. These experimental descriptions have been removed from section 3 and moved to section 2.